

# Influence of long term nitrogen limitation on lipid, protein and pigment production of *Euglena gracilis* in photoheterotrophic cultures

Marika Tossavainen[1], Usman Ilyass[1,2], Velimatti Ollilainen[3], Kalle Valkonen[1,4], Anne Ojala[1,5,6] and Martin Romantschuk[1]

[1] Faculty of Biological and Environmental Sciences, Ecosystems and Environment Research Programme, University of Helsinki, Lahti, Finland
[2] Oy Soya Ab/Jalofoods, Tammisaari, Finland
[3] Department of Food and Nutrition Sciences, University of Helsinki, Helsinki, Finland
[4] Kyrö Distillery Company, Isokyrö, Finland
[5] Faculty of Agriculture and Forestry, Institute of Atmospheric and Earth System Research (INAR)/Forest Sciences, University of Helsinki, Helsinki, Finland
[6] Faculty of Biological and Environmental Sciences, Helsinki Institute of Sustainability Science (HELSUS), University of Helsinki, Lahti, Finland

Corresponding author
Marika Tossavainen,
marika.tossavainen@helsinki.fi

## ABSTRACT

Nitrogen limitation is considered a good strategy for enhancement of algal lipid production while conversely N repletion has been shown to result in biomass rich in proteins. In this study, the influence of long-term N limitation on *Euglena gracilis* fatty acid (FA), protein, chlorophyll *a*, and carotenoid concentrations was studied in N limited cultures. Biomass composition was analyzed from three-time points from N starved late stationary phase cultures, exposed to three different initial N concentrations in the growth medium. Total lipid content increased under N limitation in ageing cultures, but the low N content and prolonged cultivation time resulted in the formation of a high proportion of saturated FAs. Furthermore, growth as well as the production of proteins, chlorophyll *a* and carotenoids were enhanced in higher N concentrations and metabolism of these cellular components stayed stable during the stationary growth phase. Our findings showed that a higher N availability and a shorter cultivation time is a good strategy for efficient *E. gracilis* biomass production, regardless of whether the produced biomass is intended for maximal recovery of polyunsaturated FAs, proteins, or photosynthetic pigments. Additionally, we showed an increase of neoxanthin, β-carotene, and diadinoxanthin as a response to higher N availability.

## INTRODUCTION

Microalgae are useful organisms in the field of biotechnology, and the main interest is currently in the production of high value substances for human and animal nutrition, cosmetics, and pharmaceuticals (*Pulz & Gross, 2004*; *Spolaore et al., 2006*). As primary

producers of long chain polyunsaturated fatty acids (LC-PUFAs) microalgae are an ecological alternative for fish oils in food, food supplements, and aquaculture feed (*Pulz & Gross, 2004*; *Harwood & Guschina, 2009*; *Van Hoestenberghe et al., 2016*) and provide a protein source with an amino acid composition comparable to other plant based proteins (*Becker, 2007*). Algal pigments are used as colorants, antioxidants, and vitamin A precursors in cosmetics and nutritional products (*Spolaore et al., 2006*; *Vílchez et al., 2011*), and anti-inflammatory, antitumor, and antibacterial properties of chlorophylls and carotenoids have been reported (*Vílchez et al., 2011*; *Da Silva Ferreira & Sant`Anna, 2017*). The potential of several microalgal strains for these applications is widely studied, but only a few strains are utilized commercially. For example, *Crypthecodinium* is a producer of LC-PUFA docosahexaenoic acid (*Pulz & Gross, 2004*), *Chlorella* and *Spirulina* are protein-rich strains used in human nutrition, *Dunaliella salina* is a natural source of β-carotene and *Haematococcus fluvialis* is used for astaxanthin production for aquaculture feed (*Spolaore et al., 2006*).

Algal growth and biomass composition is regulated by environmental factors, such as light, temperature and availability of essential nutrients, and carbon (C). Growth is enhanced under optimal light and temperature and with abundant nutrients and C. Generally, in nutrient replete growth conditions, cellular C is mostly allocated to formation of nitrogen (N) containing macromolecules, especially proteins, and nucleic acids, but also chlorophylls, amino acids, and betaine glycine (*Geider & La Roche, 2002*). However, variation in N allocation between cellular protein (59.3–96.8% of Total N) and non-protein N containing compounds is broad, and the importance of non-protein compounds tends to be greater during the exponential growth phase than in aging cultures (*Lourenço et al., 1998*). In addition, C partitioning between starch and lipids has been shown to be regulated by N availability. In N replete conditions, C is allocated to starch synthesis whereas lipid synthesis is inhibited and vice versa (*Wang et al., 2015*). In N limited growth, cellular C flow turns into non-N containing compounds, especially neutral lipids, carbohydrates, and carotenoids (*Geider & La Roche, 2002*). Cellular phosphorus (P) is mostly allocated to RNA, DNA, and phospholipids (PLs) (*Geider & La Roche, 2002*). Within the lipid fraction, growth phase and the availability of C, N, and P also influence cellular FA composition. Generally, under optimal growth conditions, cellular lipids mainly consist of membrane lipids such as PLs, which are rich in polyunsaturated FAs (PUFAs) and C-PUFAs (*Hodgson et al., 1991*). Synthesis of saturated and monounsaturated FAs (SAFAs and MUFAs), typical in storage lipids, is enhanced during the stationary growth phase under nutrient depletion (*Hodgson et al., 1991*).

Among the microalgae, *Euglena gracilis* is known as a producer of PUFAs (*Schwarzhans et al., 2015*), proteins (*Becker, 2007*), vitamins B, C, and E (*Baker et al., 1981*; *Takeyama et al., 1997*), chlorophylls *a* and *b*, several types of carotenoid pigments (*Takaichi, 2011*), and the carbohydrate paramylon (β- 1,3-glucan) (*Santek et al., 2009*). A decrease in cellular PUFA content in stationary growth phase cells in comparison to exponential growth phase cells in N deprived, photoheterotophic *E. gracilis* cultures has been shown (*García-Ferris et al., 1996*) and the protein content has been proven to decrease under N limitation (*Regnault, Piton & Calvayrac, 1990*). A decrease in chlorophyll and total carotenoid

production by *E. gracilis* was observed during a short term exposure to N deprivation (*García-Ferris et al., 1996*).

So far, production of particular cellular components of *E. gracilis* as a response to environmental factors have mostly been studied separately, and in many studies the biochemical composition is analyzed only at single time points (*Hulanicka, Erwin & Bloch, 1964*; *Rocchetta et al., 2006*). However, earlier studies have observed a transition in cellular biochemical composition between the exponential and stationary growth phases (*Regnault et al., 1995*). To date, the most thorough study of FA formation during the growth of *E. gracilis* under photoheterotrophic and heterotrophic conditions with different concentrations of glucose (C source) and proteose peptone (N source) is provided in the study by *Schwarzhans et al. (2015)*. However, *Schwarzhans et al. (2015)* did not analyze the cellular nutritional status of *E. gracilis*, and it is thus unclear whether C or N limited the growth.

For utilization of microalgae biomass as such or for maximizing production of specific compounds it is important to determine the influence of growth conditions on biomass composition. Since N limitation is often considered the most critical factor regulating cellular metabolism, this study aimed to measure the time-dependent influence of long-term N deprivation on growth and biomass composition of *E. gracilis* in photoheterotrophic growth conditions and to investigate the influence of N limitation on cellular metabolism during the stationary phase. Biomass N and C accumulation, total lipid, protein, chlorophyll *a*, and carotenoid contents as well as the FA profile of *E. gracilis* were analyzed at three time-points of late stationary phase. Main carotenoids were identified and quantified at the end of the cultivation. Our first hypothesis was that higher N content in the growth medium boosts N and C uptake and thus the growth of *E. gracilis*, and that exposure to N limitation increases lipid production and decreases protein production. Second, we hypothesized that the proportion of PUFAs is higher in biomass grown under high initial N concentrations and it decreases as a function of time. Third, higher N concentrations were hypothesized to favor chlorophyll *a* production while carotenoid production was assumed to be independent of N concentration since the carotenoids, acting in light harvest and photoprotection, do not contain N (*Christaki et al., 2012*).

## MATERIAL AND METHODS

### Strain, medium, and culturing methods

*Euglena gracilis* (CCAP 1224/5Z) was cultivated in autoclaved (120 °C, two bar, 1 h) modified Hutner medium (*Takeyama et al., 1997*) with the following modifications; L-Glutamic acid was not used, the amount of glucose was reduced to five g $L^{-1}$, $(NH_4)_2SO_4$ was added as an extra N source, $CaCO_3$ was replaced with $CaCl_2$ (0.2 g $L^{-1}$), and the amount of vitamin $B_{12}$ was doubled from the original to 0.02 mg $L^{-1}$. Three different N levels were used in the experiments: 0.0, 0.2, and 0.5 g $L^{-1}$ of $(NH_4)_2SO_4$ (hereafter called low N=LN, medium N=MN, and high N=HN treatments). The composition of the cultivation medium and the modified trace element solution are given in Table 1. The initial ammonium nitrogen ($NH_4$-N) concentrations in the treatments were 42.5, and 84.9 and 148.5 mg $L^{-1}$, respectively. N limitation, rather than P or C, during the stationary

**Table 1 Composition of culture medium.**

| Reagent | (g L$^{-1}$) |
|---|---|
| Glucose | 5 |
| $(NH_4)_2SO_4$ | 0.0–0.5[a] |
| $KH_2PO_4$ | 0.4 |
| $(NH_4)_2HPO_4$ | 0.2 |
| $MgSO_4 \cdot 7H_2O$ | 0.5 |
| $CaCl_2$ | 0.2 |
| $H_3BO_3$ | 0.0144 |
| Vitamin B1 | 0.0025 |
| Vitamin B12 | 0.00002 |
| [b]Trace element stock solution (in g 100 mL$^{-1}$ MQ-water) | |
| $ZnSO_4 \cdot 7H_2O$ | 4.4 |
| $MnSO_4 \cdot H_2O$ | 1.16 |
| $NaMoO_4 \cdot 2H_2O$ | 0.3 |
| $CuSO_4 \cdot 5H_2O$ | 0.32 |
| $CoCl_2 \cdot 6H_2O$ | 0.28 |
| [c]Fe-solution (in g 100 mL$^{-1}$ MQ water) | |
| $(NH_4)_2SO_4Fe(SO_4)_2 \cdot 6H_2O$ | 1.14 |
| EDTA | 1.0 |

**Notes:**
[a] 0.0, 0.2, and 0.5 g L$^{-1}$.
[b, c] one mL of stock solutions was added to one L of base medium.
LN, low N; MN, medium N; HN, high N treatments.

growth phase, was confirmed by using high P containing medium and both glucose and $CO_2$ as a C source.

Culturing with three replicates was done in a growth chamber (SANYO growth cabinet MLR-350 H; 294L; SANYO Electric Co. Ltd, Osaka, Japan) in two L borosilicate bottles with a cultivation volume of 1.6 L. A total of 10 mL of algal seed culture with DW 2.7 g L$^{-1}$ was used as an inoculant. Cultivation bottles were equipped with aeration, degassing, and harvesting pipes. The light and dark cycle was 16:8, light intensity 170 μmol m$^{-2}$ s$^{-1}$ (Li-Cor 190R Quantum Sensor and LI-1400 Light Sensor Logger; Li-Cor, Lincoln, NE, USA), and temperature 25 °C. Cultures were fed with 2% $CO_2$ (99.8%) in moist air (0.5 L min$^{-1}$) during the light period. For mixing the supplied gas, compressed air (10 L min$^{-1}$, Hailea 318 air compressor) and $CO_2$ (0.2 L min$^{-1}$) were pumped via flasks half filled with distilled $H_2O$, and from the gaseous phase of the flask, the mixture of moist air and $CO_2$ was supplied to the cultivation bottles trough PTFE membrane filters (Acro®37 TF Vent).

## Sampling and growth determination

Biomass growth was followed as DW, with samples taken twice a week. DW was determined as described by *Tredici & Zittelli (1998)* from samples filtered onto pre-dried (105 °C, overnight) glass fiber filters (GF/C 47; Whatman, Maidstone, UK). Specific growth rates (μ, d$^{-1}$) during the exponential growth phase (days 0–5) were calculated using the equation $μ = Ln(DW_1/DW_0)/(t_1 - t_0)$, where $DW_0$ and $DW_1$ are the biomass DWs

at the beginning ($t_0$) and end ($t_1$) of the exponential growth. The supernatants were collected for analysis of $NH_4$-N in the cultivation medium. Exponential growth ceased after 5 days and samples for C, N, FA, and total carotenoid and chlorophyll *a* analysis were taken from the late stationary growth phase on cultivation days 14, 16, and 19 and for HPLC analysis of carotenoid pigments on cultivation day 19. Biomass was collected on day 19 by centrifugation (Heraeus Multifuge 1S-R, Kendro Laboratory Products, Osterode, Germany) (3,000 rpm, 4 °C, 15 min). Algal pellets for C and N analyses were stored at −20 °C and for lipid and pigment analysis at −70 °C. Before analysis, biomass pellets were freeze-dried (Christy® Alpha 1-4; B. Braun Biotech International, Melsungen, Germany). Approximately 100 mg of dried biomass was used for C and N analyses and lipid extraction, and 40 mg for pigment extraction.

## Analytical methods

Ammonium nitrogen in the growth medium was analyzed using Hach Lange Kits (Hach Lange, Düsseldorf, Germany) and a DR 2800TM spectrophotometer (Hach Lange, Germany). C and N content of the biomass was analyzed with a Leco CNS-2000 analyzer (Leco Corporation, St Joseph, MI, USA). Lipids were extracted according to the method described by *Parrish (1999)* and modified by *Natunen et al. (2017)*. Methylation was carried out with the modified method (*Natunen et al., 2017*) of *Christie & Han (2010)*. FAs were analyzed with GC–MS (GC/MS-QP 2010 Ultra SYSTEM; Shimadzu, Canby, OR, USA) equipped with an autosampler (AOC-20 s; Shimadzu, Kyoto, Japan) and the operating software (GCMS solution, Version 2.6) using a DB-23 capillary column (Agilent Technologies, Santa Clara, CA, USA). The temperature program for GC–MS analysis was set as described earlier (*Natunen et al., 2017*; *Tossavainen et al., 2017*). FAs were identified using retention times and mass spectra of FAs in FAME standard solution (Supelco™ 37 Component FAME Mix; Supelco, Bellefonte, PA, USA). For quantification, a quantitative FAME standard mix was prepared in four concentrations and standard curves were made for each FAME. FAs were quantified with internal standard method using deuterated octadecanoic acid (C18:0-d3) (Larodan Fine Chemicals, Solna, Sweden) as an internal standard (*Natunen et al., 2017*). Total FA (TFA) content was calculated as a sum of quantified FAs. Biomass protein content was calculated by multiplying cellular N content by 4.78, which is the average conversion factor for microalgae (*Lourenço et al., 2004*).

Pigments were extracted by accelerated solvent extraction (ASE-350; Dionex, Sunnyvale, CA, USA) (110 °C, 20 min) using acetone as an extraction solvent. After extraction, acetone was evaporated under nitrogen flow, and samples were then dissolved in 10 mL of MeOH for HPLC analysis. For total carotenoid and chlorophyll analysis, one mL of MeOH extract was evaporated, and the pigments were dissolved in EtOH and filtered (PTFE syringe filter, 0.2 μm; VWR International, Radnor, PA, USA) before analysis. Total carotenoid and chlorophyll *a* concentrations were measured spectrophotometrically (UV–Vis spectrophotometer, UV-2401PC; Shimadzu, Suzhou, China) using wavelengths 450 nm for carotenoids and 665 nm for chlorophyll *a*. Total carotenoid concentration was calculated using Beer–Lambert's law and a specific absorption coefficient of 2,620 ($A^{1\%}$ $cm^{-1}$) for β-carotene in ethanol (*Rodriguez-Amaya,*

*2011*). Chlorophyll *a* concentration was calculated using absorption coefficient 84 (L g$^{-1}$ cm$^{-1}$) and the standard protocol (SFS 5772).

For HPLC analysis of carotenoid pigments, two separate standard mixes with five (first mix) and four (second mix) concentrations for identification and quantification were prepared. The first contained fucoxanthin, neoxanthin, astaxanthin, zeaxanthin, cantaxanthin, β-carotene (Sigma-Aldrich Chemie GmbH, Schnelldorf, Germany) and lutein (CaroteNature GmbH, Ostermundigen, Switzerland) (Fig. S1), and the second included violaxanthin, diadinoxanthin, diatoxanthin, alloxanthin, myxoxantophyll, and echinenone (DHI, Hørsholm, Denmark) (Fig. S2). Trans-β-apo-8′-carotenal (Sigma-Aldrich Chemie Gmbh, Germany) was added both to the standard mix and to the samples to confirm stability of retention times. Carotenoids were analyzed from MeOH extracts with HPLC (Prominence liquid chromatograph, LC-20AT; Shimadzu) using YMC carotenoid C30 column (250 × 4.6 mml.D.) (YMC America, Inc., Allentown, PA, USA), and the detection wavelength 450 nm (Prominence UV/Vis detector; Shimadzu, Suzhou, China). Essentially the analysis was performed as described in the column manufacturer's instructions: Elution solvent A for MeOH:MTBE:H$_2$O was prepared according to instructions (81:15:4 vol), and elution solvent B was slightly modified (16:80:4 vol). The flow rate was one mL min$^{-1}$ and the running time in analysis (100% A to 100% B) was 55 min. Concentrations were quantified using the external standard method. All the extraction and preparation steps were carried out in dim light to avoid deterioration of pigments.

A high definition mass spectrometer (Synapt G2-Si Q-Tof; Waters, Milford, MS, USA) equipped with APCI interface in positive mode was used to characterize diadinoxanthin in *E. gracilis* samples. The instrument settings were as follows: mass range 50–2,000 amu, corona current seven μA, probe and source temperatures 400 and 120 °C, desolvation gas 880 L/h, trap collision energy 30V. For accurate mass measurement the instrument was calibrated with a mixture of sodium iodide and Ultramark standard material. Proper mass calibration was considered to be <3 ppm. Leucine enkephaline ((M + H)$^+$ = 556.2766 amu; Waters, Milford, MS, USA) served as a lock mass calibrant.

### Statistics

Influence of time and N treatment on N and C accumulation, growth (biomass DW), protein, TFA, PUFA, MUFA, chlorophyll *a*, and carotenoid, contents were analyzed using repeated measures ANOVA. Tukey's test was used as a post hoc test. Because of heterogeneity in variances (Levene statistics, test of homogeneity of variances), statistics for SAFA and LC-PUFA contents were performed with the non-parametric Friedman's test. A significance level of $P < 0.05$ was used in all tests. Results from non-parametric tests were Bonferroni corrected. All statistical analyses were done with SPSS (Version 24; IBM, New York City, NY, USA).

## RESULTS

### Growth and C:N ratio

N availability regulated biomass growth of *E. gracilis*. Higher N content in the cultivation medium boosted growth, and differences in biomass production between LN, MN,

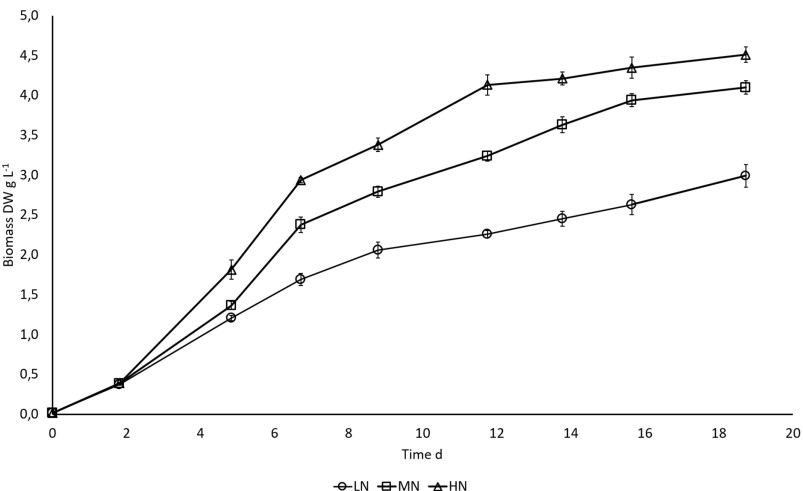

**Figure 1 Biomass growth (DW) in *E. gracilis* cultures.** LN, low N (○); MN, medium N (□); HN, high N (Δ). Mean ± SE, $n = 3$, error bars are not visible. Differences in biomass DW were statistically significant ($P < 0.05$) in different N treatments.

and HN treatments started to appear on day 5 when exponential growth ceased (Fig. 1). Growth rates during the exponential growth phase were 0.88 ± 0.01, 0.91 ± 0.01, and 0.97 ± 0.02 d$^{-1}$ in LN, MN, and HN treatments, respectively. A slow increase in biomass DW continued until the end of cultivation in all cultures, but the increase was significant ($P < 0.05$) only in MN and HN cultures. Biomass yield was highest in HN and lowest in LN cultivation ($P < 0.05$) at all three sampling points when the samples for biochemical analysis were taken. At the end of the cultivation, biomass yields in HN, MN, and LN treatments were 4.5, 4.1, and 2.9 g L$^{-1}$, respectively.

In all cultures, NH$_4$-N was rapidly removed from the cultivation medium, apparently as a result of uptake by the algae (Fig. 2). NH$_4$-N was taken up almost completely after five cultivation days in LN (91.1%) and MN (97.1%) treatments and on day 7 in HN (98.2%) treatment.

The proportions of N and C in stationary phase biomass reflected the amount of added NH$_4$-N in the medium, that is, the highest initial NH$_4$-N addition (HN treatment) resulted in higher biomass N and C concentrations ($P < 0.05$), whereas there was no significant difference between MN and LN treatments ($P > 0.05$). In all cultures biomass N content decreased and C content increased slightly, but not significantly ($P > 0.05$) between days 14 and 19, when the cultures had reached the stationary phase (Table 2). The proportion of N and C in DW after 14, 16, and 19 days of cultivation were 2.0–2.1% and 45.5–47.8% in LN, 2.3–2.5% and 47.0–47.9% in MN and 3.7–3.9% and 47.9–48.8% in HN treatments, respectively (Table 2). Based on the molar C:N ratio in algal biomass (under optimal growth conditions C:N ratio = 6.6; *Redfield, 1958*; *Geider & La Roche, 2002*) all the cultures were N limited at the time samples for C:N analysis were taken. In all cultures the C:N ratio increased toward the end of cultivation. In the LN treatment, the C:N ratio was 25 on day 14 and increased to 28 in the last samples. In MN and HN treatments, C:N ratios were 22 and 14 on the

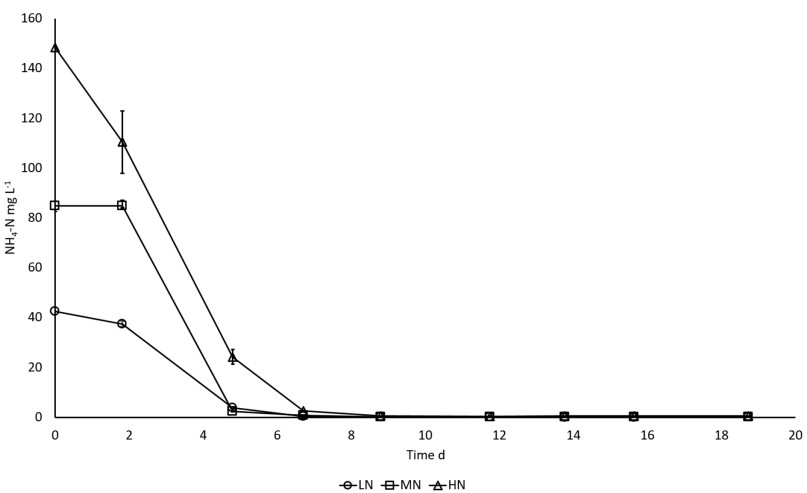

**Figure 2 NH$_4$-N removal in *E. gracilis* cultures grown under different initial N concentrations.** LN, low N (○); MN, medium N (□); HN, high N (Δ). Mean ± SE, $n = 3$, smallest error bars are not visible.

**Table 2 Proportions of C and N and molar C:N ratio in the biomass under different N treatments in the late stationary phase cultures on cultivation days 14, 16, and 19.**

|  | Day 14 | | | Day 16 | | | Day 19 | | |
|---|---|---|---|---|---|---|---|---|---|
|  | LN | MN | HN | LN | MN | HN | LN | MN | HN |
| C (%) | 45.45 ± 0.56 | 47.02 ± 0.49 | 47.90 ± 0.36 | 46.74 ± 0.64 | 47.53 ± 0.13 | 48.63 ± 0.21 | 47.81 ± 0.50 | 47.88 ± 0.34 | 48.81 ± 0.65 |
| N (%) | 2.1 ± 0.11 | 2.53 ± 0.13 | 3.88 ± 0.13 | 2.0.7 ± 0.08 | 2.43 ± 0.08 | 3.81 ± 0.11 | 1.98 ± 0.08 | 2.3 ± 0.12 | 3.66 ± 0.11 |
| C:N | 25 | 22 | 14 | 26 | 23 | 15 | 28 | 24 | 16 |

Notes:
Values for C and N are mean ± SE ($n = 3$).
LN, low N; MN medium N; HN, high N.

first sampling day and increased to 24 and 16, respectively, towards the end of the cultivation (Table 2).

## Biochemical composition

In all N-treatments, TFA content of *E. gracilis* increased toward the end of the cultivation ($P < 0.05$) (Fig. 3). Although TFA contents were always highest in the late growth phase on day 19, that is, 121, 123, and 99.5 mg g$^{-1}$ (12.1%, 12.3%, and 9.95% of DW) in LN, MN, and HN treatments, respectively, the differences between cultures were not statistically significant ($P > 0.05$).

Culture age, as well as N-treatment, influenced the degree of FA unsaturation. Generally, the prolonged cultivation time resulted in higher SAFA and lowered PUFA contents. The SAFA content increased in MN and HN treatments ($P < 0.05$) and the PUFA content decreased in MN treatment ($P < 0.05$) (Table 3). The MUFA and LC-PUFA contents remained stable ($P > 0.05$) in all cultures until the end of the cultivation (Table 3). Influence of different N treatments on proportions of SAFAs and LC-PUFAs were significant only when comparing HN and LN treatments. The contents of SAFAs were lower (44.4–52.9%) and LC-PUFAs higher (26.1–29.3) in HN treatment than in LN

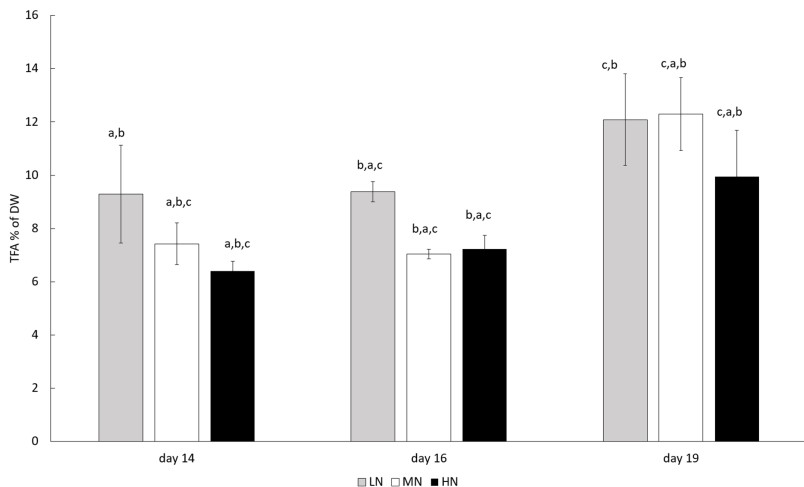

**Figure 3  TFA contents in different N treatments in the late stationary phase cultures on days 14, 16, and 19.** LN, low N (gray column); MN, medium N (white column); HN, high N (black column). Mean ± SE, $n = 3$. Statistically significant ($P < 0.05$) differences in TFA contents (% of DW) during the cultivation are shown (a = day 14, b = day 16, c = day 19). Differences in LN, MN, and HN treatments were not statistically significant ($P > 0.05$).

treatment (SAFAs 68.1–71.5% and LC-PUFAs 18.2–19.2%) ($P < 0.05$), whereas the differences were not significant between LN and MN or MN and HN treatments ($P > 0.05$) (Table 3). The PUFA content was highest in HN treatments (13.4–18.3%), and MUFA content was lower in the cultures from LN treatment (6.2–6.5%) than in other cultures ($P < 0.05$). The main FAs in all cultures were C14:0 and C16:0 SAFAs. The most abundant PUFA in all cultures was α-linolenic acid (ALA) and the most abundant LC-PUFAs were arachidonic acid (ARA) and eicosapentaenoic acid (EPA) (Table 3). In addition, *E. gracilis* is known to produce C16:4 PUFA (*Shibata et al., 2018*; *Tossavainen et al., 2018*), but since it was not included in our standards, it was excluded from the analysis. The nutritional status of cultures clearly influenced FA metabolism. The high overall C:N ratio in LN treatment and the increase in C:N ratio in all cultures during the experiment resulted in a higher content of C14:0 and a lowered C16:0 content. Lower C:N ratio favored the synthesis of PUFAs and LC-PUFAs, especially ALA, ARA, and EPA.

In each N-treatment biomass protein content was stable ($P > 0.05$) during the stationary phase, but in the HN treatment the content was higher than in other cultures ($P < 0.05$). Protein contents in HN treatment varied between 175.1 and 185.6 mg g$^{-1}$ (17.51–18.56% of DW) whereas the protein contents in LN and MN treatments were 100.2–101.9 mg g$^{-1}$ (10.02–10.19% of DW) and 109.9–120.9 mg g$^{-1}$ (10.99–12.09% of DW) (Fig. 4).

In all treatments, the biomass total carotenoid and chlorophyll *a* concentrations were stable during the stationary phase (Figs. 5A and 5B) ($P > 0.05$). Chlorophyll *a* and total carotenoid concentrations were higher in HN treatment (6,554–8,250 and 2,975–3,109 μg g$^{-1}$) than in MN (3,595–4,638 and 1,734–1,960 μg g$^{-1}$) or LN (2,493–3,260 and 873–1,337 μg g$^{-1}$) treatments ($P < 0.05$) (Figs. 5A and 5B). Diadinoxanthin (Fig. S2), β-carotene and neoxanthin (Fig. S1) were identified as the most abundant carotenoid
**Table 3** Percentage of FAMEs and sum of SAFAs, MUFAs, PUFAs, and LC-PUFAs in the late stationary phase cultures on days 14, 16, and 19.

| FAME | Day 14 | | | Day 16 | | | Day 19 | | |
|---|---|---|---|---|---|---|---|---|---|
| | LN (%) | MN (%) | HN (%) | LN (%) | MN (%) | HN (%) | LN (%) | MN (%) | HN (%) |
| C12:0 | 2.4 ± 0.4 | 1.4 ± 0.1 | 0.9 ± 0.0 | 2.5 ± 0.1 | 2.3 ± 0.2 | 2.1 ± 0.3 | 2.2 ± 0.1 | 2.6 ± 0.2 | 1.9 ± 0.5 |
| C13:0 | 12.4 ± 1.0 | 5.7 ± 0.8 | 3.7 ± 0.1 | 13.4 ± 0.3 | 6.0 ± 0.9 | 5.4 ± 1.8 | 14.9 ± 0.7 | 7.7 ± 0.6 | 7.2 ± 2.5 |
| C14:0 | 28.0 ± 0.6 | 21.5 ± 0.9 | 14.8 ± 0.2 | 28.9 ± 0.6 | 23.6 ± 0.3 | 19.2 ± 2.3 | 30.7 ± 0.1 | 25.2 ± 0.0 | 21.0 ± 3.3 |
| C15:0 | 7.1 ± 0.1 | 3.2 ± 0.4 | 2.4 ± 0.2 | 7.7 ± 0.1 | 3.1 ± 0.4 | 2.5 ± 0.6 | 8.7 ± 0.1 | 3.5 ± 0.5 | 3.3 ± 0.7 |
| C16:0 | 17.3 ± 0.3 | 21.3 ± 0.5 | 20.9 ± 0.1 | 15.8 ± 0.2 | 20.9 ± 0.7 | 19.8 ± 1.3 | 14.3 ± 0.3 | 19.4 ± 0.6 | 18.7 ± 2.1 |
| C16:1 | 4.0 ± 0.1 | 4.8 ± 0.2 | 4.0 ± 0.1 | 3.9 ± 0.1 | 5.3 ± 0.2 | 4.4 ± 0.3 | 3.9 ± 0.1 | 5.2 ± 0.3 | 4.4 ± 0.5 |
| C18:0 | 0.9 ± 0.1 | 0.9 ± 0.1 | 1.7 ± 0.7 | 0.9 ± 0.0 | 1.0 ± 0.1 | 1.0 ± 0.0 | 0.8 ± 0.0 | 1.2 ± 0.4 | 0.9 ± 0.0 |
| C18:1(n-9c) | 2.5 ± 0.2 | 3.3 ± 0.3 | 3.9 ± 0.8 | 2.3 ± 0.2 | 3.2 ± 0.3 | 3.0 ± 0.1 | 2.2 ± 0.0 | 3.6 ± 0.5 | 3.1 ± 0.0 |
| C18:2(n-6c) | 2.9 ± 0.2 | 4.5 ± 0.2 | 6.6 ± 0.5 | 2.3 ± 0.2 | 3.8 ± 0.0 | 5.8 ± 1.0 | 1.9 ± 0.2 | 3.3 ± 0.1 | 5.1 ± 1.1 |
| C18:3(n-3) | 3.5 ± 0.3 | 6.4 ± 0.3 | 11.7 ± 1.7 | 3.0 ± 0.5 | 5.2 ± 0.5 | 9.9 ± 1.9 | 2.2 ± 0.3 | 3.9 ± 0.5 | 8.3 ± 2.0 |
| C20:2 | 2.7 ± 0.1 | 3.5 ± 0.2 | 3.1 ± 0.1 | 2.5 ± 0.1 | 3.2 ± 0.2 | 2.6 ± 0.0 | 2.4 ± 0.0 | 3.1 ± 0.1 | 2.6 ± 0.1 |
| C20:3(n-6) | 0.7 ± 0.1 | 1.0 ± 0.1 | 1.0 ± 0.1 | 0.7 ± 0.1 | 0.9 ± 0.1 | 0.8 ± 0.1 | 0.7 ± 0.1 | 0.9 ± 0.1 | 0.8 ± 0.1 |
| C20:4(n-6) | 5.7 ± 0.7 | 9.0 ± 0.4 | 10.4 ± 0.1 | 5.8 ± 0.3 | 8.6 ± 0.2 | 9.5 ± 0.4 | 5.5 ± 0.2 | 8.0 ± 0.1 | 8.5 ± 0.5 |
| C20:3(n-3) | 1.0 ± 0.0 | 1.6 ± 0.1 | 1.6 ± 0.0 | 1.0 ± 0.0 | 1.5 ± 0.1 | 1.6 ± 0.0 | 0.9 ± 0.0 | 1.4 ± 0.1 | 1.8 ± 0.1 |
| C20:5(n-3) | 6.1 ± 0.4 | 8.2 ± 0.4 | 9.1 ± 0.0 | 6.5 ± 0.3 | 8.2 ± 0.3 | 8.8 ± 0.2 | 6.2 ± 0.2 | 8.0 ± 0.2 | 9.1 ± 0.5 |
| C22:6(n-3) | 2.6 ± 0.2 | 3.6 ± 0.2 | 4.1 ± 0.1 | 2.7 ± 0.1 | 3.2 ± 0.2 | 3.4 ± 0.2 | 2.5 ± 0.1 | 3.0 ± 0.1 | 3.3 ± 0.3 |
| SAFA | 68.1 ± 1.7 | 54.0 ± 1.4 | 44.4 ± 1.3 | 69.2 ± 1.0 | 56.8 ± 1.0 | 50.0 ± 3.8 | 71.5 ± 0.6 | 59.7 ± 0.3 | 52.9 ± 4.9 |
| MUFA | 6.5 ±0.3 | 8.2 ± 0.3 | 8.0 ± 0.7 | 6.2 ± 0.1 | 8.5 ± 0.4 | 7.5 ± 0.2 | 6.2 ± 0.1 | 8.8 ± 0.3 | 7.6 ± 0.5 |
| PUFA | 6.4 ± 0.4 | 11.0 ± 0.4 | 18.3 ± 2.2 | 5.4 ± 0.7 | 9.1 ± 0.4 | 15.7 ± 2.9 | 4.1 ± 0.5 | 7.2 ± 0.6 | 13.4 ± 3.1 |
| LC-PUFA | 19.0 ± 1.5 | 26.8 ± 1.4 | 29.3 ± 0.2 | 19.2 ± 0.4 | 25.6 ± 0.9 | 26.8 ± 0.7 | 18.2 ± 0.2 | 24.4 ± 0.2 | 26.1 ± 1.4 |

**Note:**
Values are mean ± SE ($n = 3$).

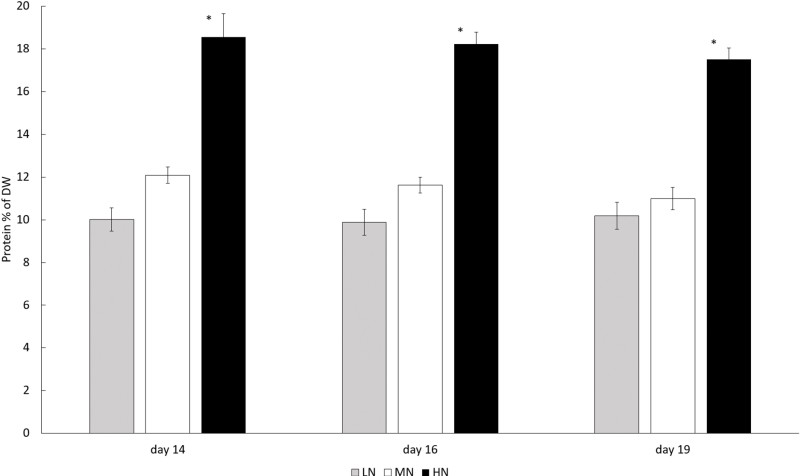

**Figure 4 Protein contents in different N treatments in the late stationary phase cultures on days 14, 16, and 19.** LN, Low N (gray column); MN, medium N (white column); HN, high N (black column). Mean ± SE, $n = 3$. Statistically significant ($P < 0.05$) differences on days 14, 16, and 19 in different N treatments are shown. *Significantly higher protein content (% of DW).

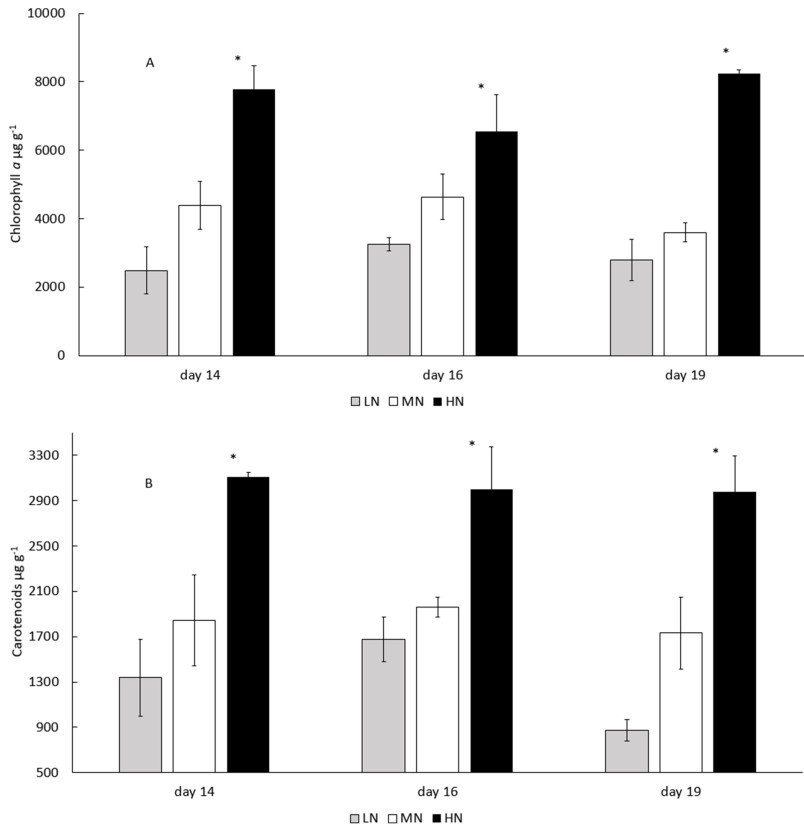

**Figure 5 Chlorophyll *a* and total catotenoid concentrations in the late stationary phase *E. gracilis* cultures on days 14, 16, and 19.** (A) Chlorophyll *a* and (B) total carotenoid concentrations (µg g$^{-1}$) in the late stationary phase cultures on days 14, 16, and 19. LN, low N (gray column); MN, medium N (white column); HN, high N (black column). Mean ± SE, $n = 3$. Statistically significant ($P < 0.05$) differences on days 14, 16, and 19 in different N treatments are shown. *Significantly higher chlorophyll *a* or carotenoid content.

pigments on the last cultivation day. Concentrations of identified carotenoids were always lowest in LN treatment and highest in HN treatment (Fig. 6). Diadinoxanthin was eluted as a front part of the double peak (Rt = 12.1 min) containing the carotenoid (m/z 583) and a green pigment (m/z 909). UV–Vis (277, 422, 445 (max), 476 nm) and mass spectra data ((M + H)$^+$ = 583.4125 amu, 3.6 ppm error, and fragment ions (m/z 565, 547, 221) confirmed the presence of diadinoxanthin (*Young & Britton, 1993*). Due to this co-elution of diadinoxanthin with a green pigment, diadinoxanthin amounts were considered only as suggestive values.

## DISCUSSION

As hypothesized, the higher N addition in HN treatment boosted N and C uptake and resulted in a higher biomass yield of *E. gracilis.* Exponential growth in all cultures ceased after 5 days, whereas the NH$_4$-N was exhausted from the medium on day 5 in LN and MN treatments and on day 7 in the HN treatment. This indicates that growth was N limited in LN and MN cultures on day 5 but in the thicker HN culture, the primary reason for growth slowing down was probably light limitation and decreased photosynthetic

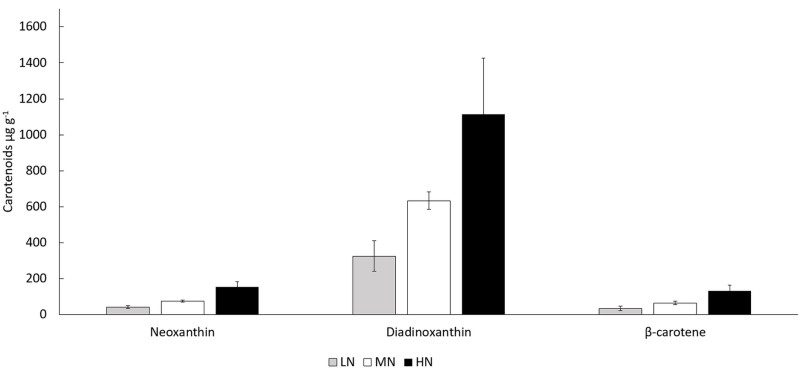

**Figure 6 Concentrations of neoxanthin, diadinoxanthin, and β-carotene in different N treatments at the end of the cultivation.** LN, low N (gray column); MN, medium N (white column); HN, high N (black column). Mean ± SE, $n = 3$.

activity. However, the high C:N ratio in stationary phase cultures shows that all cultures were N limited in late stationary phase; this was seen already on day 14 when the first samples for C:N analysis were taken. The slow assimilation of C to biomass in stationary phase cultures was not significant ($P > 0.05$). Biomass DW increased significantly only in MN and HN treatments between days 14 and 19 ($P < 0.05$). Sufficient amount of $NH_4$-N in HN culture resulted also in higher C and N contents ($P < 0.05$). Thus, the differences in biomass production between cultures can be explained by influence of initial $NH_4$-N concentrations on cellular metabolism. Continued C assimilation and growth under short-term exposure to N starvation has been shown earlier in cultures of *Isochrysis zhangjiangensis*, whereas long-term N deprivation restricted both assimilation and growth as a response to decreased photosynthetic activity (*Wang et al., 2015*). Deterioration of photosynthetic pigments in N deprived cells has been shown earlier (*Wang et al., 2015*; *Da Silva Ferreira & Sant`Anna, 2017*) and indications of this effect was also seen in this study. Additionally, C uptake of *E. gracilis* from organic substrates (*Ogbonna, Ichige & Tanaka, 2002*) allows C accumulation also under conditions restricting photosynthesis, which makes growth under N deprivation less sensitive to chlorophyll degradation.

Total FAs were the only cellular compounds, which quantitatively were influenced by prolonged incubation ($P < 0.05$), whereas the differences between treatments were not significant ($P > 0.05$). This result indicates that for TFA production of *E. gracilis*, culture age is more significant than N concentration and that under long-term N limitation, and C replete conditions, C flow turns to formation of non-N containing FAs. C allocation to lipid synthesis in C rich and N deplete conditions has also been shown earlier (*Wang et al., 2015*). Clear enhancement of lipid production in aging cultures has also been observed earlier (*Schwarzhans et al., 2015*), and here this enhancement took place regardless of initial N concentrations. Additionally, as shown earlier, significant enhancement of FA content occurs after the transition from exponential growth phase to the stationary phase (*Regnault et al., 1995*). In addition, the availability of C is essential for FA synthesis, and in C and N limited stationary phase cultures of *E. gracilis*,

C deficiency results in lower FA content than under C replete conditions, regardless of N availability (*Regnault et al., 1995*). However, under heterotrophic growth conditions, paramylon formation is enhanced, whereas lipid production is improved in photoheterotrophic growth (*Schwarzhans et al., 2015*). Earlier, high glucose content has been shown to shift metabolism of *E. gracilis* toward heterotrophy (*Schwarzhans et al., 2015*). Thus, photoheterotrophy in the low glucose medium used in this study can be assumed to be a good strategy for lipid production in N depleted growth conditions. The C availability for lipid formation is ensured under N limited conditions which restricts chlorophyll formation and thus inhibits photosynthetic C assimilation.

Algal FA composition defines the usefulness of the produced lipids for different applications. Following earlier studies (*Regnault et al., 1995*; *Schwarzhans et al., 2015*), our results showed that high N concentration and short cultivation time results in an FA composition of good nutritional quality since generally the LC-PUFA and PUFA contents were high and SAFA contents were low. The influence of time for FA saturation degree was significant ($P < 0.05$) only for PUFAs in MN and SAFAs in MN and HN treatment. This indicates that FA metabolism is more sensitive to initial N concentrations than to the length of time under N deprivation. LC-PUFAs and PUFAs are typically structural compounds in cell membranes and their proportion in total FAs is relatively high in optimal growth conditions (*Hodgson et al., 1991*). Thus, it can be assumed, that relative proportion of membrane lipids in cultures grown in HN conditions was higher than in other treatments.

Also, the high content of C14:0 SAFA in LN and MN treatments emphasizes the importance of nutritional status. C14:0 is more injurious to humans than other SAFAs, since it more efficiently elevates the blood LDL cholesterol, thus increasing the risk of cardiovascular diseases (*Dubois et al., 2007*). A slight increase of C14:0 and decrease in C16:0 SAFA contents were seen here during the growth in all cultures. A similar trend of simultaneous lowering of C16:0 content and increase of C14:0 content has been shown earlier in aging cultures (*Schwarzhans et al., 2015*). Low N and high organic C content in the growth medium induces wax ester synthesis in *E. gracilis* cells (*Regnault et al., 1995*) and the high content of SAFAs in LN treatment indicates the same. Harvesting in the late stationary phase ensures high biomass content and TFA yield. However, a cultivation of *E. gracilis* under N depletion results in an undesirable FA profile with low PUFA and high SAFA content, and is thus not a recommended method for production of lipids for food or feed applications. Furthermore, from the seventh day onward, the biomass in the HN treatment was much higher than in LN treatment. The higher biomass compensates the lower TFA content and results in similar TFA yields, but with less saturated FAs. By knowing this, we can simplify the production process of *E. gracilis* based PUFAs and LC-PUFAs; maximum yield is achieved faster allowing a short harvesting cycle. With the proposed method a long N starvation period is not needed for maximizing the yield thereby saving costs in large scale production. Results also indicate that for the optimal LC-PUFA and PUFA production, maximizing biomass production of *E. gracilis* instead of FA content is more important.

As assumed, higher N availability resulted in higher protein content ($P < 0.05$), indicating that N augmentation is a good strategy for production of protein-rich biomass for food or feed applications. However, the protein content in our cultures was low in comparison to earlier reported protein contents of 31–61% in *E. gracilis.* In general, protein content in algal biomass can vary between 6% and 63% (*Becker, 2007*). Since N is the key component in proteins, the low protein content of our cultures was a consequence of N limitation. A general response to lower cellular N content and higher C:N ratio in biomass is a decrease in protein as well as non-protein N compounds, as has been shown for several marine microalgae (*Lourenço et al., 2004*). A drastic decrease in cellular protein content of *E. gracilis* immediately after exposure to N depletion was demonstrated by *Regnault, Piton & Calvayrac (1990)*.

Chlorophyll *a* concentration increased as a response to HN treatment ($P < 0.05$), a phenomenon shown earlier (*Regnault, Piton & Calvayrac, 1990*; *Lourenço et al., 2004*). Declining chlorophyll concentrations are typical of stationary phase algal cultures (*Lourenço et al., 2004*), and since N is a structural component in chlorophyll (*Da Silva Ferreira & Sant`Anna, 2017*), this decline is probably at least partly a consequence of nutrient limitation. Chlorophyll degradation under N depletion has been observed in cultures of *E. gracilis* and *I. zhangjiangensis* (*García-Ferris et al., 1996*; *Wang et al., 2015*), but our results reveal that chlorophyll *a* concentrations are stable during the stationary phase ($P < 0.05$). Thus, we assume that degradation of chlorophyll at the end of the exponential growth phase and its stability during the late stationary growth phase might be a response to the switch from autotrophic to photoheterotrophic growth in conditions where chlorophyll formation and photosynthetic activity is inhibited. The influence of changing light intensities on increased chlorophyll *a* synthesis in HN treatment cannot, however, be completely excluded, since light limitation has been shown to boost chlorophyll synthesis (*Geider, MacIntyre & Kana, 1997*).

Against our hypothesis, in the HN treatment, the total carotenoid concentrations also increased ($P < 0.05$). Structurally, carotenoids are N free compounds, and the observed increase under HN conditions may be related to the need for N in the formation of pigment-protein complexes in the thylakoid membrane (*Takaichi, 2011*). Alternatively, production of light harvesting primary carotenoids was enhanced in the dense culture. The primary carotenoids have functions in photosynthesis, as light harvesting or photoprotecting pigments, whereas the secondary carotenoids are metabolized under stress conditions (*Christaki et al., 2012*), that is, during nutrient deficiency or high light intensity (*Grung & Liaaen-Jensen, 1993*). However, response to N as well as P limitation seems to be specific for different carotenoid pigments. For example, the content of the secondary carotenoid astaxanthin in *Haematococcus pluvialis* increased under N and P limitation, whereas the concentrations of primary carotenoids lutein and β-carotene, and the secondary carotenoid cantaxanthin decreased (*Boussiba et al., 1999*). The major carotenoids identified here (neoxanthin, diadinoxanthin, and β-carotene), have been classified as primary carotenoids of *E. sanguinea* (*Grung & Liaaen-Jensen, 1993*).

We could confirm diadinoxanthin as the most abundant carotenoid pigment in *E. gracilis,* which is following earlier findings (*Brandt & Wilhelm, 1990*;

*Schagerl, Pichler & Donabaum, 2003*; *Kato et al., 2017*). Diatoxanthin is a minor carotenoid in *E. gracilis* (*Schagerl, Pichler & Donabaum, 2003*; *Takaichi, 2011*; *Kato et al., 2017*), but it was not present in our cultures, whereas β-carotene was shown here and earlier to be one of the major carotenoids of the species (*Goodwin & Jamikorn, 1954*; *Heelis et al., 1979*; *Takaichi, 2011*). However, both diadinoxanthin and diatoxanthin are pigments syntethized in the so-called diadinoxanthin cycle, and in diatoms this cycle has importance in photoprotection (*Lepetit et al., 2010*). In high light conditions, diadinoxanthin is de-epoxidized to diatoxanthin, and in low light intensity, diatoxanthin is epoxidazed back to diadinoxanthin (*Lepetit et al., 2010*). This might explain the lack of diatoxanthin in our dense cultures. The stability of total carotenoid content during the stationary phase ($P > 0.05$) shows that in *E. gracilis*, carotenoids are not degraded when the growth conditions chance unfavorable for photosynthesis. Thus, a rapid transformation of stored diadinoxanthin to diatoxanthin can provide an excellent photoprotection system when cells are exposed to high light intensity. Lutein has also been identified as an abundant carotenoid in *E. gracilis* (*Goodwin & Jamikorn, 1954*), but this was later claimed to be incorrect, and the corresponding pigment was first identified as antheraxanthin (*Krinsky & Goldsmith, 1960*) and later as diadinoxanthin (*Heelis et al., 1979*).

Our study is the first one to reveal the influence of N limitation on carotenoid composition of *E. gracilis* on the late stationary growth phase. The response to N availability was similar for all quantified carotenoids, which is in line with earlier studies showing that concentrations of β-carotene in *H. fluvialis* (*Boussiba et al., 1999*) and diadinoxanthin in *Heterocapsa* sp. (*Latasa & Berdalet, 1994*) cultures decrease under N limitation. Knowledge of carotenoid production in *E. gracilis* is still insufficient and contradictory, and in future, the differences between genotypes or response of pigment biosynthesis to different growth conditions should be clarified. Presently, diadinoxanthin is not utilized in biotechnology. The diadinoxanthin cycle pigments are important in photoprotection and singlet oxygen scavenging in algae, and antioxidative properties and the potential applications of diadinoxanthin need further investigations.

## CONCLUSION

This study showed that long term N limitation is not a good strategy to boost lipid production of *E. gracilis* for nutritional use. Long cultivation time and strict N limitation results in higher TFA concentrations but poor FA composition with low PUFA and LC-PUFA concentrations and high SAFA content. Greater availability of N results in higher protein, chlorophyll *a*, and carotenoid concentrations. Thus, N availability is critical for the maximal production of PUFAs, LC-PUFAs, proteins and pigments, and long-term N limitation is not a recommended method for production of *E. gracilis* biomass for nutritional purposes.

## ACKNOWLEDGEMENTS

We thank our laboratory personnel Riikka Koivula and Santeri Savolainen, for helping in the laboratory work. We also thank John Allen, who checked the language of the manuscript.

## Funding

This work was supported by Business Finland (No. 40409/11, 40091/14 and 6803/31/2017) and the European Regional Development Fund (ERDF) (Häme, Päijät-Häme and Uusimaa regions, Levarbio-project, No. A71035). Additional funding was obtained from TES (Finnish Foundation for Technology Promotion, Gasum Gas Fund). The funders had no role in study design, data collection and analysis, decision to publish, or preparation of the manuscript.

## Grant Disclosures

The following grant information was disclosed by the authors:
Business Finland: 40409/11, 40091/14, and 6803/31/2017.
European Regional Development Fund (ERDF) (Häme, Päijät-Häme and Uusimaa regions, Levarbio-project): A71035.
TES (Finnish Foundation for Technology Promotion, Gasum Gas Fund).

## Competing Interests

The authors declare that they have no competing interests. K. Valkonen is a shareholder of the Kyrö Distillery Company. Usman Ilyass is employed by Oy Soya ab/Jalofoods.

## Author Contributions

- Marika Tossavainen conceived and designed the experiments, performed the experiments, analyzed the data, prepared figures and/or tables, authored or reviewed drafts of the paper, approved the final draft.
- Usman Ilyass conceived and designed the experiments, performed the experiments, authored or reviewed drafts of the paper, approved the final draft.
- Velimatti Ollilainen conceived and designed the experiments, performed the experiments, authored or reviewed drafts of the paper, approved the final draft.
- Kalle Valkonen conceived and designed the experiments, performed the experiments, authored or reviewed drafts of the paper, approved the final draft.
- Anne Ojala conceived and designed the experiments, authored or reviewed drafts of the paper, approved the final draft.
- Martin Romantschuk conceived and designed the experiments, authored or reviewed drafts of the paper, approved the final draft.

## Data Availability

Chromatograms of carotenoid pigments in standard mixes and examples of chromatograms of samples are available in Figs. S1 and S2.

The raw data are available in Datasets S1–S5. Raw data shows dry weights (Dataset S1), biomass C, N and protein contents (Dataset S2), NH4-N removals (Dataset S3), total fatty acid (TFA) concentrations, FA contents (% of DW), and percentages of individual FAs (Dataset S4). Total chlorophyll $a$ and total carotenoid concentrations, and

concentrations of neoxanthin, diadinoxanthin, and β-carotene are in Dataset S5. These values were used in statistical analysis to compare DW, biomass C, N and protein contents, total FA contents, proportion of SAFAs, MUFAs, PUFAs, and LC-PUFAs and total chlorophyll *a* and carotenoid concentrations in different N (LN, MN, and HN) treatments.

## Supplemental Information

Supplemental information for this article can be found online at http://dx.doi.org/10.7717/peerj.6624#supplemental-information.

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
