# Peer review of "Influence of long term nitrogen limitation on lipid, protein and pigment production of Euglena gracilis in photoheterotrophic cultures"

_PeerJ, doi:10.7717/peerj.6624_

## Round 0.1 · original submission · Major Revisions

Thank you for your submission. I would like to inform you that your manuscript has been recommended for publication subject to major revisions according to the given referee reports. I also found your manuscript well written and nicely presented. In addition to our referees reports, I would like to suggest these following minor changes (i) adding proper axis-label on the y-axis of Figures 1& 3 & 4& 5 (i.e. biomass, g.L-1 or etc…); (ii) axis-labels (x and y axis) and also units (x axis) in Figure 2 . Once again, thank you for submitting your manuscript to PeerJ and I look forward to receiving your revision.

Reviewer 1 ·

Basic reporting

The manuscript is well written and covers adequate background knowledge and literature beings reviewed

Experimental design

The experimental design of this study is adequate and addresses the research question raised while also testing the hypotheses made in the introduction

Validity of the findings

no comment

Additional comments

This study looked into the influence of nitrogen depletion on lipid composition and other biochemical components of Euglena gracilis. The manuscript is well written and covers adequate background knowledge and literature beings reviewed. My main concern of this study is the lack of novelty as it has been significantly established in the field of Applied Phycology that nitrogen depletion of cultures results in an increase in the total lipid and TFA content of most microalgae. Similar studies on Euglena has been reported in early work by Garcia Ferris et al., 1996 and Regnault et al., 1995. Nevertheless, an innovative component of this study is the initial growth of Euglena at different N concentration and the effect of the different N concentration during the N depletion period.
Major comments
1) The author need to better highlight the justification behind this work in terms of large scale TFA and PUFAs production. If the method used in this study had worked and subsequently scaled up, would commercial production facilities first grow the algae and then subsequently starve (deplete) the cells for TFA and PUFAs production. If yes how efficient and cost effective would this be compared to conventional algae cultivation for PUFAs and TFA.

2) As the authors have discussed in the discussion section, the enhanced production of the carotenoid diadinoxanthin during the N depletion period is the only positive outcome of this work. Is there any commercial potential/ benefits for this carotenoid? If yes it should be highlighted in the introduction and discussions section of this manuscript.

3) Line 284-286: The authors stathe that exponential growth in all cultures ceased after Day 5 when N was exhausted but in their results section state that N was only exhausted at Day 7 for HN. This is contradicting and will affect the results

Minor Comments
1) Line 299: Need of proof/reference to support the claim that C is being converted to FA.

2) Line 219-220: The authors state that the higher N content ‘boosted the growth of the algae’ but did not evaluate the specific growth rate (µ) of the algae in each treatment. I would recommend the authors to calculate the µ based on the increase in biomass of the three treatments and compare them as it would provide additional important information.

3) The manuscript needs to be proof read to include punctuation and sentence connectors such as commas and other connecting words. Many long sentences are comprised of two or more different statements which are awkwardly combined together
Example: Line 246-247
Line 232-233 and a few more

4) Line 251-253: The changes in SAFA, PUFA and MUFA needs to be better explained whether it was changes observed over time or changes between the different treatments.

5) Figures: There should be some form of indication in the figure to illustrate whether there were any significant difference among the different treatments. You can either use symbols or alphabets on top of each bar chart to highlight significant difference among the treatments

6) The abbreviations ARA, ALA and EPA needs to be first referred as to its full terminology before the use of the abbreviation for the ease of the readers not familiar with this abbreviations

7) Chlorophyll a needs to be changed to Chlorophyll a


8) The values (units) for protein and TFA should be represented as % of biomass rather than mg/g as this is the preferred reporting style of many studies and for ease of comparison with other similar studies.
Based on the questions being raised, I would only recommend the manuscript for publication only after major revision.
Thank you

Reviewer 2 ·

Basic reporting

The manuscript is within the scope of the journal and the Introduction adequately introduced.

Experimental design

1. The authors analyzed three time points of the nitrogen starved Euglena cells grown in photoheterotrophic culture. But they only analyzed the changes of compounds during stationary phase, not in other time points, such as exponential phase. To compare the value with other literature, it is very important to show that they observed similar results to previous reports or not. At this point, it is not clear whether difference between previous reports (such as Regnault et al 1995) and their results came from culture conditions (medium, light intensity and bubbling of CO2) or growth phase.

2. The authors used GC-MS for fatty acid analysis, but GC-MS is not suitable for quantitative analysis, because of difference of ionization tendency for each fatty acid. If the authors want to quantify fatty acids by GC-MS, they have to use standards for each fatty acid molecular specie or change detector to FID (flame ionization detector). If the authors can not do the above analysis, they can focus on comparison of relative amount of each fatty acid species among samples (based on the DW).

Validity of the findings

Data is not robust enough because of the above mentioned issue.

Additional comments

1. Line 66, spell out TN (total N?).
2. Line 85 and 88, the reference would be (Garcia-Ferris et al., 1996).
3. Line 97, “through” might be “thorough”.
4. Line 120, 1bar would be 2 bars.
5. Line 127, explanation of NH4-N might be needed.
6. Line 191 and 193 “Schimadzu” would be “Shimadzu”.
7. Line 210 and 247, what is TFA (total fatty acid?)?
8. Line 275, “douple” would be “double”.

---

## Round 0.2 · Minor Revisions

Thank you for your re-submission and your efforts to fulfill both referees’ and my previous comments. As also suggested by our referees, I would like to recommend "minor revisions" before its final publication in PeerJ. I look forward to receiving the final version of your manuscript. Best wishes.

Reviewer 1 ·

Basic reporting

The manuscript is well revised and written. The authors have incorporated most suggestions and answered question asked previously

Experimental design

The experiment is planned and carried out adequately.

Validity of the findings

The results obtained through is robust and statistically sound. The small error margins help confirm this

Additional comments

The authors should mention and discuss in text regarding the statistical difference of parameter they measured (i.e. growth, yield, FA) rather than only including this information in the Figure Caption.


Line 300-301: Could you include the day this happened? “However, the high C: N ratio in stationary phase cultures shows that all cultures were N limited in late stationary phase”.

Reviewer 2 ·

Basic reporting

no comment

Experimental design

no comment

Validity of the findings

no comment

Additional comments

This is the second submission of the manuscript “Influence of long term nitrogen limitation on lipid, protein and pigment production of Euglena gracilis in photoheterotrophic cultures” by Tossavainen et al. The authors responded to the reviewers comments mostly. This reviewer has few comments as follow:

1. The authors analyzed three time points of the nitrogen starved Euglena cells grown in photoheterotrophic culture. If this reviewer understand correct, the authors only compared samples from stationary phase, because biomass of the culture is the most abundant in stationary phase and they tried to find out the best time point to have better FAs for nutritional purpose. If this is true, describe it somewhere (maybe in Discussion).

2. The authors used GC-MS for fatty acid analysis, and made standard curves of each FAs using FAME standards for quantification analysis. If the authors did so, it should be written in the text.

3. The FAME standards the authors used does not include 4,7,10,13-hexadecatetraenoic acid (16:4), which is a major FA in Euglena gracilis (more than 10% of total FA). The authors’ analysis is probably missing this PUFA. (see Hulanicka et al 1964 JBC, Constantopoulos et al 1967 JBC, Shibata et al 2018 Front Plant Sci).

4. On line 255, the authors stated that TFA amount is the lowest in the cells grown with HN treatment, even this cells have more chlorophyll a and carotenoids. This reviewer is wondering that HN treatment make the biomass of the cells higher than the cells with other treatments because of higher photosynthetic activity. For photosynthesis, development of thylakoid membrane is necessary and it requires membrane lipid synthesis, which lead to accumulation of FA (attached to the membrane lipids). In this manuscript, C content was not analyzed in detail, so it would be interesting to see how the C content changes (especially photosynthetic activity and accumulation of paramylon) by the treatments they used, in future.

Miscellaneous things:
1. Again, line 85 and 88, the reference would be (Garcia-Ferris et al., 1996), because this paper is written by four authors.

2. Again, line 120, 1bar would be 2 bars, otherwise it is not autoclave.

3. Line 124, 0.02 g L-1 would be 0.02 mg L-1. 0.02 g L-1 is the concentration for the stock solution. If the value authors described is correct, it is 2,000 times higher than the original medium (also check Table 1).

4. Line 198, which is “elution solvent A”?

5. Line 301, there is double periods.

---

## Round 0.3 · accepted · Accept

Thank you for your re-submission and your additional efforts to fulfill both of our referees’ follow-up comments. I am pleased to inform you that your manuscript, entitled as " Influence of long term nitrogen limitation on lipid, protein and pigment production of Euglena gracilis in photoheterotrophic cultures” has been accepted for publication in PeerJ. Many thanks for your contribution, and we look forward to seeing more of your work in the future. Best wishes.

#